# Linearizing Vision Transformer with Test-Time Training

Yining Li [* 1]   Dongchen Han [* 1]   Zeyu Liu [* 1]   Hanyi Wang [1]   Yulin Wang [1]   Gao Huang [1]

## Abstract

While linear-complexity attention mechanisms offer a promising alternative to Softmax attention for overcoming the quadratic bottleneck, training such models from scratch remains prohibitively expensive. Inheriting weights from pretrained Transformers provides an appealing shortcut, yet the fundamental representational gap between Softmax and linear attention prevents effective weight transfer. In this work, we address this conversion challenge from two perspectives: architectural alignment and representational alignment. We identify Test-Time Training (TTT) as a linear-complexity architecture whose two-layer dynamic formulation is structurally aligned with Softmax attention, enabling direct inheritance of pretrained attention weights. To further align representational properties, including key shift-invariance and locality, we introduce key instance normalization and a lightweight locality enhancement module. We validate our approach by linearizing Stable Diffusion 3.5 and introduce SD3.5-T$^5$ (**T**ransformer **T**o **T**est **T**ime **T**raining). With only 1 hour of fine-tuning on 4×H20 GPUs, SD3.5-T$^5$ achieves comparable text-to-image quality to the fine-tuned Softmax model, while accelerating inference by 1.32× and 1.47× at 1K and 2K resolutions. Code is available at this URL.

## 1. Introduction

Transformer architectures have fundamentally reshaped modern machine learning, serving as the backbone for state-of-the-art systems across computer vision, natural language processing, and multimodal reasoning. The success of these models is largely attributed to the Softmax attention mechanism, which enables flexible modeling of long-range dependencies through pairwise token interactions. However, this

expressive power comes at a prohibitive cost: the time and memory complexity grows quadratically with the sequence length $N$. This limitation severely restricts scalability, particularly for applications involving high-resolution images or long-context generation.

To address this bottleneck, extensive research has focused on developing linear-complexity attention mechanisms (Katharopoulos et al., 2020). While training linear models from scratch is feasible, a more practical yet challenging goal is to *linearize* existing large pre-trained Transformers (e.g., DiT, Stable Diffusion) to accelerate inference without incurring the colossal cost of full retraining. Despite numerous attempts, efficiently converting a pre-trained Softmax attention model into a linear-complexity one remains an open challenge. The core difficulty lies in the significant disparity in representational spaces between Softmax attention and standard linear attention. Concretely, the weights learned in the Softmax paradigm can hardly be mapped to the linear attention space, necessitating extensive retraining or complex distillation strategies. As a result, existing conversion methods often inherit only partial pretrained weights (e.g., MLP layers) and knowledge, failing to achieve linearization within a few fine-tuning steps.

In this work, we tackle this conversion challenge from two complementary perspectives: *structure alignment and representation alignment*.

Firstly, we aim to find a linear-complexity architecture that shares a representational space structurally compatible with Softmax attention, thereby supporting direct weight inheritance and fast adaptation. We identify Test-Time Training (TTT) (Sun et al., 2025) as the ideal candidate for this task. TTT reformulates sequence modeling as an online learning problem, where an inner model is dynamically updated by the input sequence. Crucially, TTT achieves linear complexity because its hidden state, i.e., the weights of the inner model, has a fixed size regardless of the sequence length $N$. Our key insight is that, unlike linear kernels, the non-linear inner model of TTT (for example, a two-layer MLP) possesses the capacity to fit the complex representational space of Softmax attention. This structural alignment allows TTT to fully inherit pre-trained Softmax weights and naturally adapt to the pre-trained non-linear manifold, combining the efficiency of linear models with the expressiveness of

---

*Equal contribution [1]Tsinghua University, Beijing, China. Correspondence to: Gao Huang <gaohuang@tsinghua.edu.cn>.

*Proceedings of the 43$^{rd}$ International Conference on Machine Learning*, Seoul, South Korea. PMLR 306, 2026. Copyright 2026 by the author(s).

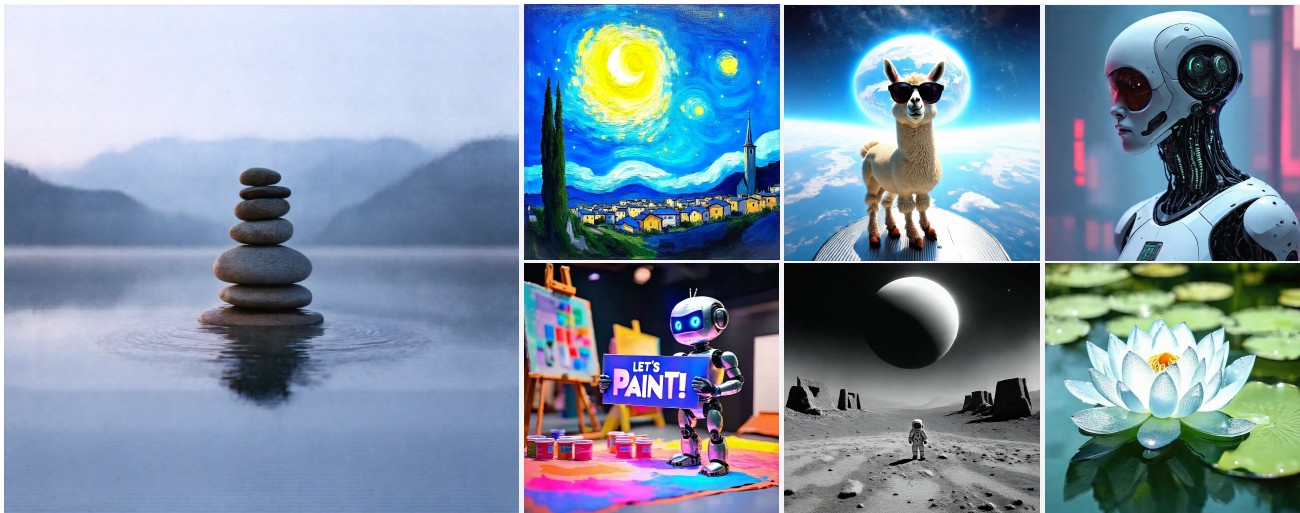

*Figure 1.* Left: 2K image generated by SD3.5-T$^5$; right: 1K images generated by SD3.5-T$^5$.

Softmax attention.

While structural compatibility opens the possibility for adaptation, the linearized models need also to emulate and align with the learned representational properties in pretrained Transformers to enable fast and accurate conversion. We identify two such properties that significantly impact linearization. First, Softmax attention is shift-invariant with respect to keys $K$ and implicitly operates on normalized attention scores. Consequently, keys $K$ learned by pretrained Transformers tend to exhibit a strong bias and are not zero-centered, which can impede rapid adaptation. Second, Softmax attention naturally learns strong local representations, a property that many linear-complexity models fail to represent. To bridge these gaps, we (i) apply Instance Normalization to key representations to mitigate bias and emulate Softmax's normalization, and (ii) introduce a lightweight locality-enhancement module to restore local representational power. These components align TTT's representational behavior with the pretrained Softmax model, enabling rapid adaptation with minimal training cost.

We validate our approach by linearizing state-of-the-art vision Transformers and diffusion Transformers. On DiT-XL/2, we achieve comparable performance to the original model with only *8 epochs* of fine-tuning (merely *0.57%* of the original training steps) after weight inheritance, bypassing the need for distillation or complex operator activation. We further demonstrate the effectiveness of our method on Stable Diffusion 3.5, where our method restores nearly full original performance with only 3000 fine-tuning steps (around 1 hour on 4× H20), delivering *1.32×* and *1.47×* speedups at 1024 and 2048 resolutions.

Our main contributions are summarized as follows:

1. We identify Test-Time Training (TTT) as a linear-complexity architecture that is structurally compatible with Softmax attention. This key insight enables full weight inheritance from pre-trained Softmax attention to a linear-complexity architecture, without requiring complex distillation or other training strategies.

2. We reveal two critical representational properties of pretrained Softmax Transformers that significantly impact linearization: key shift-invariance and strong locality bias. Based on these analyses, we introduce instance normalization on key representations and a lightweight locality-enhancement module, aligning the behavior of TTT with that of pretrained Softmax models and enabling data-efficient adaptation.

3. We validate our approach in image classification and image generation tasks. On DiT-XL/2, our method achieves comparable performance with only 8 epochs of fine-tuning (0.57% of the original training steps), without distillation or complex operator activation. On Stable Diffusion 3.5, we recover nearly full performance with only 3K fine-tuning steps.

## 2. Related Work

**Efficient Vision Transformers.** While Vision Transformers (Dosovitskiy et al., 2020; Touvron et al., 2021a;b) have achieved remarkable success, their quadratic attention complexity $\mathcal{O}(N^2)$ limits scalability to long visual sequences. Prior works address this through local attention (Liu et al., 2021; Hassani et al., 2023; Dong et al., 2022), sparse patterns (Wang et al., 2021; Xia et al., 2022; Zhu et al., 2023), circulant paradigms (Han et al., 2026a), or linear-complexity formulations (Katharopoulos et al., 2020; Gu & Dao, 2024; Liu et al., 2024; Yang et al., 2024a;b; Han et al., 2023). In

this work, we focus on TTT-based linear attention approach for efficient visual modeling.

**Test-Time Training (TTT).** TTT (Sun et al., 2025) introduces a novel paradigm for sequence modeling by reformulating attention as an online learning process, where the model adapts its hidden states through gradient-based updates at inference time. This formulation enables linear-complexity $\mathcal{O}(N)$ sequence modeling while maintaining expressive power. Recent works have extended TTT to various domains, including language modeling (Zhang et al., 2025b; Behrouz et al., 2025), video generation (Dalal et al., 2025), and 3D reconstruction (Chen et al., 2025), demonstrating its versatility and scalability. Recent ViT[3] (Han et al., 2026b) explores design principles for TTT in vision backbones, laying a solid foundation for applying TTT to visual tasks. In this work, our focus is how to effectively transform pretrained Transformers into TTT models with minimal performance loss.

**Linearizing Pretrained Transformers.** Despite the emergence of numerous linear-complexity architectures, retraining large models from scratch with these new architectures remains prohibitively expensive. This has motivated a growing body of work on efficiently converting pretrained Softmax Transformers into linear-complexity alternatives. In the language domain, Hedgehog and LoLCATs (Zhang et al., 2024; 2025a) introduce learnable activations for queries and keys to approximate Softmax behavior, while other works distill pretrained Transformers into state-space models (Bick et al., 2024; Wang et al., 2024; 2025b). In the vision domain, CLEAR (Liu et al., 2026) linearizes pretrained models by restricting Softmax attention to local windows. LiT (Wang et al., 2025a) achieves rapid conversion from Softmax to linear attention by inheriting only MLP weights. Diffusion Grafting (Chandrasegaran et al., 2026) systematically studies module grafting strategies and proposes a two-stage quick-fine-tuning paradigm. These works often combine architectural changes with additional training strategies such as distillation, activation alignment, or multi-stage conversion. Our work is orthogonal to these training strategies: we focus on the architectural perspective and identify a linear-complexity structure that is inherently compatible with Softmax attention, enabling a fast and minimalist linearization approach based on full weight inheritance. The training strategies mentioned above are largely complementary to our approach and may provide further gains when combined with our architecture.

## 3. Preliminary

**Softmax Attention.** Given query $q \in \mathbb{R}^d$, keys $K \in \mathbb{R}^{N \times d}$, and values $V \in \mathbb{R}^{N \times d}$, Softmax attention computes:

$$\mathrm{Attn}(q, K, V) = \mathrm{Softmax}\left(\frac{qK^\top}{\sqrt{d}}\right)V \qquad (1)$$

**Linear Attention.** Linear attention replaces Softmax with a kernel function $\phi(\cdot)$:

$$\mathrm{LinearAttn}(q, K, V) = \frac{\phi(q) \cdot (\phi(K)^\top V)}{\phi(q) \cdot \phi(K)^\top \mathbf{1}} \qquad (2)$$

**Test-Time Training (TTT).** TTT maintains an inner model $f_W$ that is updated during inference, containing learnable parameters $W$ (optimized during training) and fast weights $\Delta$ (computed per sequence via gradient descent on a self-supervised loss). The TTT framework offers two key choices of freedom:(1) inner model architecture: The structure of $f_W$ can vary from a single linear layer, to a two-layer MLP, or even convolutional networks, providing different capacity-efficiency trade-offs.(2) Self-supervised loss: The inner optimization objective can take different forms, such as inner product loss or reconstruction loss (L2). Notably, when $f_W$ is a single linear layer and the loss is the inner product loss, TTT degenerates to linear attention. We adopt TTT with a MLP inner model $f_W(x) = \sigma(xW_1)W_2$ and L2 reconstruction loss as an example:

$$\mathcal{L}_{\mathrm{inner}} = \sum_i \|f_W(k_i) - v_i\|^2 \qquad (3)$$

The output for query $q$ is computed as $\mathrm{TTT}(q) = \sigma(qW_1')W_2'$, where $W_1' = W_1 - \Delta_1$, $W_2' = W_2 - \Delta_2$, and $\Delta_1 = \nabla_{W_1}\mathcal{L}_{\mathrm{inner}}$, $\Delta_2 = \nabla_{W_2}\mathcal{L}_{\mathrm{inner}}$. Following standard TTT practice, $W_1$ and $W_2$ are randomly initialized learnable parameters and are optimized end-to-end, while the fast-weight updates $\Delta_1, \Delta_2$ are computed from the current sequence. Intuitively, Softmax attention constructs a dynamic network losslessly from cached $K, V$, whereas TTT learns to compress the $K, V$ cache into a compact inner model on which the query operates.

## 4. Method

### 4.1. Model Structure Alignment

A central challenge in transforming pretrained Softmax attention to efficient linear attention is that existing linear attention approximations struggle to recover the performance of Softmax models, even after fine-tuning. We argue that the root cause lies in a fundamental mismatch in representational capacity: Softmax attention implicitly implements a *two-layer dynamic MLP with nonlinearity*, while standard linear attention is limited to a *single-layer dynamic linear transformation*. This gap in expressive power makes it difficult for linear attention to faithfully approximate the representation space learned by Softmax attention, resulting in poor weight inheritance and slow convergence.

*Softmax Attention as a Two-Layer Dynamic MLP.* From the query perspective, Softmax attention can be rewritten as:

$$\mathrm{Attn}(q, K, V) = \sigma(qW_1^{\mathrm{dyn}})W_2^{\mathrm{dyn}} \qquad (4)$$

where $W_1^{\mathrm{dyn}} = K^\top \in \mathbb{R}^{d\times N}$, $W_2^{\mathrm{dyn}} = V \in \mathbb{R}^{N\times d}$, and $\sigma(\cdot)$ is row-wise Softmax. Thus, each query passes through a two-layer MLP with a nonlinear activation, dynamically constructed from the $K$ and $V$ matrices, enabling rich aggregation of global context.

In contrast, standard linear attention (e.g., via kernel-based approximations) can be expressed as:

$$\mathrm{LinearAttn}(q) = \phi(q) \cdot \underbrace{(\phi(K)^\top V)}_{\text{single-layer } W^{\mathrm{dyn}}} \qquad (5)$$

where $\phi(K)^\top V$ forms a single $d\times d$ dynamic weight matrix. This single-layer linear structure lacks the intermediate nonlinearity required to closely match the representational space of Softmax attention and provides no additional learnable parameters for adaptation during transfer.

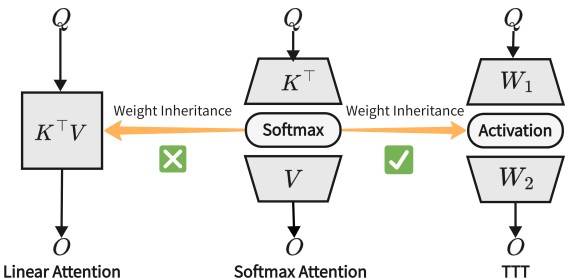

*Figure 2.* Structural similarity between Softmax Attention and two-layer TTT enables direct weight inheritance and fast adaptation.

*TTT Bridges the Gap with Matching Structure and Flexible Nonlinearity.* To overcome these limitations, we adopt Test-Time Training (TTT) layers, which replace traditional attention with a lightweight inner model that operates dynamically on the KV cache at test time. Crucially, TTT allows flexible design of this inner model. For a two-layer TTT, the output is:

$$\mathrm{TTT}(q) = \sigma(qW_1')W_2', \qquad (6)$$

where $W_1' = W_1 - \nabla_{W_1}\mathcal{L}_{\mathrm{inner}}$ and $W_2' = W_2 - \nabla_{W_2}\mathcal{L}_{\mathrm{inner}}$ consist learnable parameters $W_1$, $W_2$. This formulation exhibits a *direct structural correspondence* with Softmax attention (Eq. 4): both apply a two-layer transformation with an intermediate nonlinearity, as shown in figure 2. This alignment enables effective weight inheritance from pretrained Softmax models , while the learnable parameters of inner model allow TTT to rapidly adapt and better fit the original Softmax representation space.

**Empirical Validation.** We validate our analysis through controlled experiments, inheriting all pretrained weights and fine-tuning for 30 epochs (Table 1). Concretely, all original Transformer parameters, including the Q/K/V/O projections, MLPs, normalization layers, and embedding/projection layers, are inherited from the pretrained Softmax model. Only

*Table 1.* Comparison of architectures under weight inheritance. All linear-complexity models in the table employ InstanceNorm (see §4.2) to stabilize fine-tuning after weight inheritance.

| Model | New Params | Freeze | FT | FLOPs |
|---|---|---|---|---|
| Softmax | – | 72.05 | – | 1.25G |
| Linear Attn | 0 | 3.71 | 63.30 | 1.13G |
| Linear + $\mathrm{Proj}_{QK}$ | 0.3M | 24.39 | 66.23 | 1.19G |
| TTT-1Layer-Gate | 0.3M | 61.95 | 67.59 | 1.25G |
| TTT-2Layer | 0.3M | 65.98 | 68.14 | 1.25G |
| TTT-3Layer | 0.5M | 67.09 | 68.93 | 1.37G |
| TTT-SwiGLU | 0.5M | **67.33** | **69.25** | 1.34G |

the newly introduced TTT-specific inner parameters are newly initialized and trained with a larger learning rate. We consider two protocols: Freeze (train only new parameters) and FT (full fine-tuning). $\mathrm{Proj}_{QK}$ denotes learnable projections applied to Q, K before the activation function, a practice introduced by Hedgehog (Zhang et al., 2024) for attention transfer. See Appendix for details. Key observations are:

- *The structural gap between linear and Softmax attention is too large for effective weight inheritance, even with additional learnable parameters.* It is shown that Vanilla linear attention achieves only 3.71 under the Freeze protocol. Even with additional learnable projections ($\mathrm{Proj}_{QK}$), the Freeze performance only improves to 24.39—still far below acceptable levels. This confirms the structural gap between linear and Softmax attention is too large for efficient weight inheritance.

- *TTT's nonlinear inner model can effectively approximate the original Softmax attention, and performance improves as the nonlinearity increases.* It is shown that TTT variants demonstrate strong performance recovering when freezing the pretrained weights. Moreover, performance consistently improves with increased nonlinearity (increased inner model's depth). This trend validates that the nonlinearity of TTT's inner model is crucial for bridging the gap with Softmax attention.

Based on these findings, we adopt TTT-SwiGLU as our default architecture, which combines two-layer structural alignment with enhanced expressiveness via gated activation. We argue that two-layer nonlinearity is adequate for approximating Softmax attention, as TTT-SwiGLU slightly outperforms TTT-3Layer (67.33% vs. 67.09%) with equal parameters. To note, the final choice of inner model is different from ViT[3] (Han et al., 2026b) due to different settings.

### 4.2. Representation Shift-Invariance Alignment

In the preceding section, we have identified a linear-complexity structure that closely matches Softmax attention

from the perspective of structure alignment. However, from the representation space standpoint, Softmax attention still exhibits certain distinctive features. In what follows, we aim to align these specific properties to better approximate the representation space of Softmax attention, focusing on two aspects: (1) *shift-invariant property*, and (2) *locality*.

In this part, we focus on shift-invariant property of Softmax attention. We find although TTT-SwiGLU demonstrates strong structural compatibility with Softmax attention, directly fine-tuning after weight inheritance leads to numerical instability—training quickly diverges to NaN. To deal with this , we further delve into gaps between Softmax attention and TTT in representation space. We have the observation below:

*Softmax is Shift-Invariant, TTT is Not.* Softmax attention is invariant to constant shifts in keys. Given query $q$ and keys $K = [k_1, k_2, \ldots, k_N]^\top$, for any shift vector $\delta \in \mathbb{R}^d$, the attention weights remain unchanged:

$$\text{Softmax}\left([q^\top(k_1 + \delta), q^\top(k_2 + \delta), \ldots, q^\top(k_N + \delta)]\right)$$
$$= \text{Softmax}\left([q^\top k_1, q^\top k_2, \ldots, q^\top k_N]\right) = \text{Softmax}(qK^\top) \tag{7}$$

since Softmax normalization absorbs the constant term $q^\top\delta$ subtracted from all logits.

In contrast, TTT lacks this shift-invariance. Taking TTT-MLP as an example, we consider the scaled inner-product loss $\mathcal{L}_t(k_t) = -v_t^\top f_W(k_t)$, $\quad f_W(k) = W_2(\sigma(W_1 k))$.

The gradient with respect to $W_1$ without and with key shift:

$$\nabla_{W_1}\mathcal{L}_t(k_t) = -\left[W_2^\top v_t \odot \sigma'(W_1 k_t)\right] \cdot k_t^\top,$$
$$\nabla_{W_1}\mathcal{L}_t(k_t + \delta) = -\left[W_2^\top v_t \odot \sigma'(W_1(k_t + \delta))\right] \cdot (k_t + \delta)^\top \tag{8}$$

By applying a first-order Taylor expansion to the activation derivative term,

$$\sigma'(W_1(k_t + \delta)) \approx \sigma'(W_1 k_t) + \sigma''(W_1 k_t) \odot (W_1\delta), \tag{9}$$

The shifted gradient can be decomposed by orders of $\delta$:

$$\nabla_{W_1}\mathcal{L}_t(k_t + \delta) \approx -\Bigg\{ \underbrace{\left[W_2^\top v_t \odot \sigma'(W_1 k_t)\right] \cdot k_t^\top}_{\mathcal{O}(\delta^0)}$$
$$+ \underbrace{\left[W_2^\top v_t \odot \sigma'(W_1 k_t)\right] \cdot \delta^\top}_{\mathcal{O}(\delta^1)}$$
$$+ \underbrace{\left[W_2^\top v_t \odot \sigma''(W_1 k_t) \odot (W_1\delta)\right] \cdot k_t^\top}_{\mathcal{O}(\delta^1)}$$
$$+ \underbrace{\left[W_2^\top v_t \odot \sigma''(W_1 k_t) \odot (W_1\delta)\right] \cdot \delta^\top}_{\mathcal{O}(\delta^2)} \Bigg\}$$
$$\nabla_{W_1}\mathcal{L}_t(k_t) \neq \nabla_{W_1}\mathcal{L}_t(k_t + \delta) \tag{10}$$

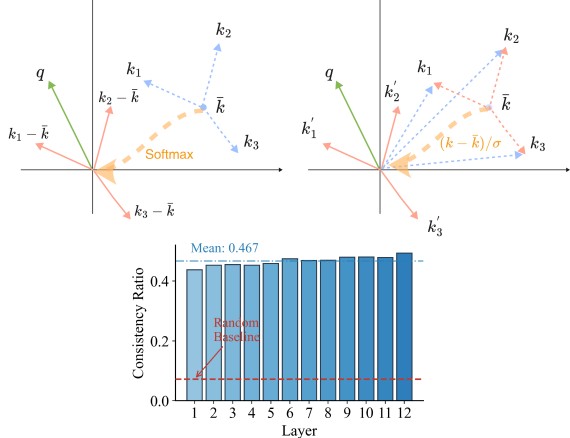

*Figure 3.* **Top:** Softmax absorbs key shifts while TTT does not. Our method recenters the keys when inheriting pretrained weights. **Bottom:** Distribution of key shift ratio across 5K images: pretrained ViT exhibits ratio $\approx 0.5$, indicating substantial key bias, while random initialization yields ratio $\approx 0.07$.

The first and second-order terms introduced by the shift are absent in the centered case; when accumulated across online TTT updates, they can amplify the inner gradients and lead to gradient explosion. This mismatch in representation space can influence the weight inheriting process. As shown in figure 3(Top), When inheriting from pretrained Softmax attention, keys may carry arbitrary biases that were previously absorbed by Softmax's shift-invariance. Without explicit handling, the inherited keys lack a well-centered initialization for TTT's optimization, which undermines training stability.

To verify whether such key shifts exist in pretrained Softmax models, we define a ratio to measure their magnitude:

$$\text{ratio} = \frac{\|\bar{k}\|_2}{\frac{1}{N}\sum_{i=1}^{N}\|k_i\|_2}, \quad \text{where } \bar{k} = \frac{1}{N}\sum_{i=1}^{N}k_i\,, N = 196 \tag{11}$$

This ratio compares the norm of the mean key (capturing the shift) against the average norm of individual keys. A ratio close to 0 indicates well-centered keys, while a ratio closer to 1 indicates larger shift. We extract keys from 5K images using a pretrained ViT and compute the average ratio. As shown in Figure 3(bottom), pretrained keys exhibit a ratio of approximately 0.5, indicating systematic key bias, while randomly initialized keys yield only 0.07. This significant shift in key distribution makes TTT lack a well-centralized initialization, resulting in training instability. To remedy this, we propose normalizing keys across instances to stabilize the training process.

**Normalizing keys with Instance Normalization.** We propose normalizing keys using InstanceNorm (Ulyanov et al., 2016) across the sequence dimension before TTT computa-

*Table 2.* Effect of normalization strategies on training stability after weight inheritance. Removing the standard-deviation scaling from InstanceNorm barely affects performance, while removing mean subtraction leads to immediate instability.

| Normalization | Stable | Acc |
|---|---|---|
| None | ✗ | 0.37 |
| RMSNorm | ✗ | 57.38 |
| LayerNorm | ✗ | 57.25 |
| **InstanceNorm (Ours)** | ✓ | **71.19** |
| InstanceNorm w/o division-by-std | ✓ | 71.15 |
| InstanceNorm w/o mean subtraction | ✗ | 51.43 |

tion:

$$\hat{k}_i = \frac{k_i - \bar{k}}{\sqrt{\frac{1}{N}\sum_{j=1}^{N}(k_j - \bar{k})^2 + \varepsilon}}, \quad \bar{k} = \frac{1}{N}\sum_{j=1}^{N}k_j \quad (12)$$

This normalization centers the keys to emulate Softmax's shift-invariance, thereby better approximating the representation space of Softmax attention and enabling more smooth weight transfer.

**Empirical Validation.** As shown in Table 2, without InstanceNorm, training diverges immediately. In contrast, Instance Normalization ensures stable fine-tuning as it performs centering over tokens, directly matching Softmax attention's shift-invariance, further bridging the gap between representation space of Softmax attention and TTT. And We compare other commonly-used normalization. Layer-Norm (Ba et al., 2016), and RMSNorm (Zhang & Sennrich, 2019) all fail—as they operate at the token level and fail to remove global key's shifts across the sequence.

Also, to test our hypothesis regarding the importance of removing the key shift, we conduct an ablation study on InstanceNorm. First, we remove the division-by-std operation in key InstanceNorm while keeping the mean subtraction. Second, we remove the mean-subtraction operation while preserving the division-by-std. Our experiments show that removing the division-by-std barely affects the accuracy, with the Top-1 accuracy changing only slightly from 71.19% to 71.15%. However, removing the mean subtraction immediately causes NaN during fine-tuning. These results confirm that removing the key shift is necessary during inherited-weight training.

### 4.3. Representation Locality Alignment

Locality is another important feature of Softmax attention in representation space, which many linear attention variants lack (Han et al., 2023; 2024). However, analyzing locality in TTT is non-trivial since it lacks explicit attention scores from QK multiplication. We first propose a method to analyze implicit attention patterns, then introduce a convolution-based locality enhancement tailored for TTT.

Unlike Softmax or linear attention, TTT does not compute explicit attention scores $QK^\top$. To analyze its locality in a unified manner, we propose computing *implicit attention scores* via gradient-based attribution: for each output token $o_i$, we measure how much each value token $v_j$ contributes to the output. Formally, we define the implicit attention score as:

$$A_{\text{implicit}}(i, j) = \frac{\partial o_i}{\partial v_j} \quad (13)$$

This gradient magnitude reflects the contribution of each value to the output, serving as a proxy for attention scores. Applied to Softmax attention, it reduces to standard attention score. Visualizing these implicit scores (Figure 4), we find that TTT exhibits weaker local attention patterns compared to Softmax attention, suggesting the need for explicit locality enhancement to match the representation space.

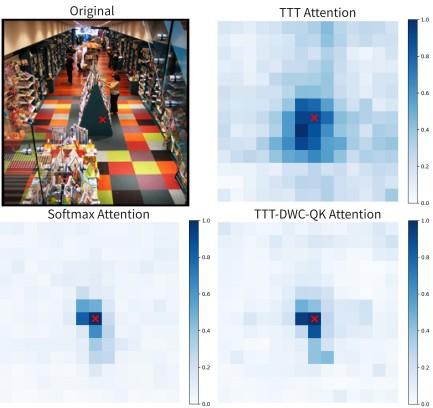

*Figure 4.* Visualizations of implicit attention scores. Softmax attention exhibits strong local bias. While TTT yield meaningful attention distributions, it focuses more on global modeling. $\text{DWC}_{QK}$ enhance the locality.

**Depthwise Convolution on Queries and Keys.** We propose enhancing locality by applying depthwise convolutions (DWC) to queries and keys before TTT computation:

$$\hat{q} = q + \text{DWC}(q), \quad \hat{k} = k + \text{DWC}(k) \quad (14)$$

This design offers two benefits: (1) it directly injects locality into the QK representations; (2) within TTT's inner learning objective $L(f_W(k), v)$, the convolved keys effectively allow a local window of keys to jointly predict each value, expanding the receptive field and improving expressiveness.

**Empirical Validation.** As shown in Figure 4, the proposed DWC module enhances the locality of TTT, further aligning with the property of Softmax attention. And we compare several locality enhancement strategies in Table 3: (1) adding convolutional positional encoding to input $x + \text{CPE}(x)$; (2) applying DWC to values $x + \text{DWC}(v)$; (3) our proposed DWC on queries and keys ($\text{DWC}_{QK}$). Results

*Table 3.* Comparison of locality enhancement strategies for TTT.

| Method | Acc | #Params | FLOPs |
|---|---|---|---|
| TTT (no locality) | 69.25 | 6.2M | 1.34G |
| + CPE($x$) | 69.64 | 6.2M | 1.34G |
| + DWC($v$) | 70.47 | 6.2M | 1.34G |
| + DWC$_{QK}$ (Ours) | **71.19** | 6.2M | 1.34G |
| + DWC$_{QK}$ + NAT3 | 71.67 | 6.2M | 1.36G |
| + DWC$_{QK}$ + NAT5 | 72.06 | 6.2M | 1.39G |

show that DWC$_{QK}$ achieves the best performance, validating our design choice. Additionally, TTT can be optionally combined with Neighborhood Attention (NAT) (Hassani et al., 2023),which computes attention within a local window:

$$\text{Output} = \frac{1}{2}\,\text{TTT}(Q, K, V; \mathcal{W}) + \frac{1}{2}\,\text{NAT}(Q, K, V) \tag{15}$$

where both branches share the same Q, K, V projections. We emphasize that unlike prior works that rely on local attention for compatibility, our method achieves full weight inheritance independently; NAT is a complementary enhancement, not a necessity.

# 5. Experiments

So far, we have finalized the roadmap. Through systematic exploration on image classification tasks, we have identified an architecture well-suited for weight inheritance. To demonstrate the generalizability of our approach, we conduct experiments not only on image classification but also on large-scale image generation models, including DiT-XL/2 and SD3.5-Medium.

## 5.1. Image Classification

We evaluate on ImageNet-1K (Russakovsky et al., 2015), a large-scale benchmark containing 1.2M training images across 1,000 categories. Our implementation is based on the official DeiT (Touvron et al., 2021a) codebase. To align with the DiT architecture used in our generation experiments, we remove the class token and adopt global average pooling; see Appendix for more details. We first train a Softmax baseline under the above configuration, then inherit all pretrained weights and convert to the TTT-DWC$_{QK}$ architecture. We denote it as T$^5$ (**T**ransformer **T**o **T**est-**T**ime **T**raining), and the model with NAT is referred to as T$^{5+}$. The converted model is fine-tuned for 30 epochs using the default learning rate. As shown in Table 4, T$^5$ rapidly recovers the baseline performance.

Table 5 further compares different methods of linearizing vision transformer. For a fair comparison focusing on architectural compatibility, we only re-implement their architectural components within our framework: 30 epochs of fine-tuning

*Table 4.* Image classification results on ImageNet-1K. DeiT$^†$ baseline reimplemented with global average pooling.

| Model | Epochs | Ratio | Top-1 | #Params | FLOPs |
|---|---|---|---|---|---|
| DeiT-T$^†$ | 300 | 100% | 72.05 | 5.7M | 1.25G |
| T$^5$-T | 30 | 10% | 71.19 | 6.2M | 1.34G |
| T$^{5+}$-T | 30 | 10% | 72.06 | 6.2M | 1.39G |
| DeiT-S$^†$ | 300 | 100% | 80.24 | 22M | 4.59G |
| T$^5$-S | 30 | 10% | 79.00 | 23M | 4.77G |
| T$^{5+}$-S | 30 | 10% | 79.64 | 23M | 4.86G |

*Table 5.* ImageNet-1K results of different efficient attention variants on DeiT-Tiny under 30-epoch training.

| Method | Acc | #Params | FLOPs |
|---|---|---|---|
| Linear | 63.30 | 5.7M | 1.13G |
| LiT | 69.52 | 5.7M | 1.14G |
| CLEAR | 68.66 | 5.7M | 1.11G |
| Hedgehog | NaN | 5.8M | 1.25G |
| T$^5$ | 71.19 | 6.2M | 1.34G |

on ImageNet without additional training techniques. Note that their training methods can also be applied to our architecture for further improvement. All methods inherit the full weights of DeiT, except that LiT does not inherit the attention weights but inherits the MLP weights, following their paper. The results show that the proposed T$^5$ architecture consistently outperforms standard linear attention, LiT, CLEAR, and Hedgehog under the same setting. Also, as shown in Figure 5, the computation efficiency becomes increasingly favorable at higher resolutions.

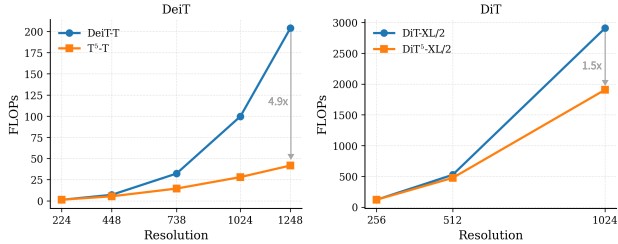

*Figure 5.* FLOPs versus resolution on DeiT (left) and DiT (right). The efficiency advantage of T$^5$ becomes more pronounced as the sequence length grows.

## 5.2. Class-Conditional Image Generation

**Setup.** Based on our designed architecture, we instantiate our method on 256×256 class-conditional image generation task. Following DiT (Peebles & Xie, 2023), we use the ImageNet-1K dataset(Russakovsky et al., 2015). Evaluation metrics include FID-50K (Heusel et al., 2017), Inception-Score (Salimans et al., 2016). We inherit all weights from the pretrained DiT-XL/2 checkpoint, directly replacing attention blocks with our TTT blocks. Notably, our approach requires no knowledge distillation and no operator activa-

tion stage. We follow all hyperparameters from DiT, except that the learning rate for TTT-specific parameters $(W_1, W_2)$ is multiplied by $2\times$ to accelerate convergence.

**Architecture.** Specifically, we replace all attention blocks with our proposed TTT blocks using the simplified TTT-2layer (MLP structure) configuration with depthwise convolution on Q, K. We do not use TTT-SwiGLU as 2-layer MLP inner model is good enough for image generation and introduces fewer parameters. The TTT inner model's loss is set as L2 loss. To further incorporate local inductive bias, we incorporate NAT with a window size of 5. We denote the model as DiT[5].

**Results.** We evaluate our method on class-conditional ImageNet generation. At $256 \times 256$ resolution, we train with a batch size of 128 for 80K steps, equivalent to 40K steps under the original DiT configuration. We compare various baselines, including U-Net based methods (ADM (Dhariwal & Nichol, 2021), CDM (Ho et al., 2022), RIN (Jabri et al., 2023), LDM (Rombach et al., 2022)), Transformer-based diffusion models (DiT (Peebles & Xie, 2023), SiT (Ma et al., 2024), Mask-GIT (Chang et al., 2022) SimpleDiff (Hoogeboom et al., 2023)), state-space models (DiM (Teng et al., 2024), DiffuSSM (Yan et al., 2024)), and recent methods of linearizing diffusion transformer (LiT (Wang et al., 2025a), Hyena-X/Y (Chandrasegaran et al., 2026)). As shown in Table 6, our method achieves an FID of **2.48** with only **0.57%** of DiT's training cost, demonstrating highly competitive performance. We attribute this efficiency to the architectural compatibility between our approach and Softmax attention, which enables full weight inheritance from pretrained models for rapid convergence. In contrast, methods such as LiT (Wang et al., 2025a) and Diffusion Grafting (Chandrasegaran et al., 2026) only inherit MLP weights, discarding the valuable knowledge in attention layers. Notably, our method requires neither distillation nor specialized operator initialization, making it both simple and efficient. As shown in Figure 5, the computational advantage of DiT[5] becomes more pronounced as the resolution increases.

### 5.3. Text-to-Image Generation

**Stable Diffusion 3.5-Medium.** To validate the generalizability of our approach, we further apply our weight inheritance strategy to Stable Diffusion 3.5-Medium (Esser et al., 2024), which poses significant computational challenges due to its requirement for processing long sequences and its substantial training overhead.

**Setup.** To linearize SD3.5-Medium, we replace approximately 50% of the transformer blocks with our proposed TTT blocks (same architecture as DiT[5] except without NAT), including 13 image-token self-attention blocks and the last 5 single attention blocks. We initialize from pre-

*Table 6.* Comparison on class-conditional ImageNet $256 \times 256$. The reported epochs for DiT[5] denote additional fine-tuning after inheriting the pretrained DiT checkpoint. [†]: Two-stage training: 14 blocks initialized with 8K samples (200 epochs each, $\approx 17.5$ ImageNet-equivalent epochs) + fine-tuning on 10% data for 50K steps ($\approx 10$ epochs).

| Model | Epochs | FT-Ratio | FID↓ | IS↑ | Replace | Distill |
|---|---|---|---|---|---|---|
| DiM-L | - | - | 2.64 | - | - | - |
| DiM-H | - | - | 2.40 | - | - | - |
| DiffuSSM-XL-G | - | - | 2.28 | 259.1 | - | - |
| ADM | - | - | 10.94 | 101.0 | - | - |
| CDM | - | - | 4.88 | 158.7 | - | - |
| RIN | - | - | 3.42 | 182.0 | - | - |
| LDM-4-G | - | - | 3.60 | 247.7 | - | - |
| SimpleDiff (U-Net) | - | - | 3.76 | 171.6 | - | - |
| SimpleDiff (U-ViT) | - | - | 2.77 | 211.8 | - | - |
| Mask-GIT | - | - | 6.18 | 182.1 | - | - |
| SiT-XL | 1400 | - | 2.06 | 277.5 | - | - |
| DiT-XL/2 | 1400 | - | 2.27 | 278.2 | - | - |
| LiT-XL/2 | 280 | 20% | 2.32 | 265.2 | 100% | ✓ |
| Hyena-X | $\sim 28$[†] | 2% | 2.74 | 273.30 | 50% | ✗ |
| Hyena-Y | $\sim 28$[†] | 2% | 2.72 | 273.37 | 50% | ✗ |
| DiT[5]-XL/2 | **8** | **0.57%** | 2.48 | 267.9 | 100% | ✗ |

trained SD3.5-Medium weights and fine-tune on open-source Flux-generated images at $1024 \times 1024$ resolution (Le, 2025) with a batch size of 64 for 3,000 steps, denoted as SD3.5-T[5]. The entire training takes approximately 1 hour on 4 NVIDIA H20 GPUs. For a fair comparison after fine-tuning, we also report variant SD3.5 fine-tuned with the identical 3,000-step procedure while keeping Softmax attention, called SD3.5-FT. We evaluate image quality on DPG-Bench (Hu et al., 2024) and GenEval (Ghosh et al., 2023). See appendix for more experimental details.

**Results.** As shown in Table 7, SD3.5-T[5] preserves generation quality while using a linear-complexity architecture and requiring only minimal fine-tuning. It outperforms the original SD3.5-Medium on DPG-Bench (84.43 vs. 83.83) and improves GenEval from 0.66 to 0.69. Under the same fine-tuning budget, SD3.5-T[5] remains comparable to SD3.5-FT, achieving 84.43 vs. 82.74 on DPG-Bench and 0.69 vs. 0.70 on GenEval. Meanwhile, our approach yields a substantial latency reduction, as illustrated in Figure 6. At $1024 \times 1024$ resolution, SD3.5-T[5] reduces latency from 25 s to 19 s, achieving a $1.32\times$ speedup. The efficiency gains become more pronounced at higher resolutions, reaching a $1.47\times$ speedup at $2048 \times 2048$, indicating the advantages of our linear-complexity architecture. To the best of our knowledge, SD3.5-T[5] is the first step toward a TTT-based text-to-image generation model. Notably, the entire process is remarkably straightforward, requiring no distillation or specialized operator initialization. By simply inheriting all weights and fine-tuning for just one hour, we obtain the SD3.5-T[5] model with comparable performance and speedup. Generated samples are shown in Figures 1.

*Table 7.* Quantitative comparison between SD3.5-Medium ([Esser et al., 2024](#)), SD3.5-FT, and SD3.5-T$^5$ on DPG-Bench and GenEval benchmarks. We also report latency (seconds) at different resolutions. Our linearized model achieves comparable performance with significant efficiency gains.

| Method | DPG-Bench | | | | | | GenEval | | | | | | | Latency (s) | |
|---|---|---|---|---|---|---|---|---|---|---|---|---|---|---|---|
| | Global | Entity | Attribute | Relation | Other | Overall | Single | Two | Count | Color | Position | Color Attri. | Overall | 1024 | 2048 |
| SD3.5-Medium | 85.04 | 88.81 | 89.92 | 91.61 | 82.91 | 83.83 | 0.99 | 0.82 | 0.58 | 0.85 | 0.23 | 0.52 | 0.66 | 25 | 231 |
| SD3.5-FT | 82.90 | 89.29 | 87.54 | 92.40 | 84.90 | 82.74 | 0.98 | 0.88 | 0.71 | 0.84 | 0.27 | 0.53 | 0.70 | 25 | 231 |
| SD3.5-T$^5$ | 91.51 | 90.03 | 87.84 | 92.17 | 87.13 | 84.43 | 1.00 | 0.88 | 0.58 | 0.88 | 0.25 | 0.55 | 0.69 | 19 | 157 |
| $\Delta$ vs. SD3.5-FT | +8.61 | +0.74 | +0.30 | -0.23 | +2.23 | +1.69 | +0.02 | 0.00 | -0.13 | +0.04 | -0.02 | +0.02 | -0.01 | 1.32$\times$ | 1.47$\times$ |

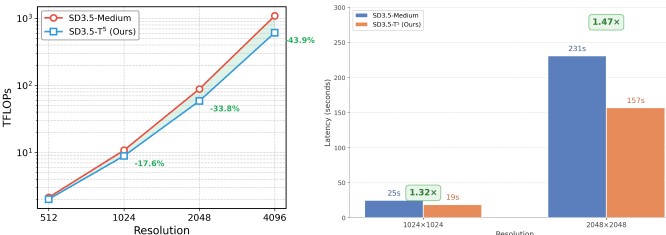

*Figure 6.* Comparison between SD3.5-Medium and SD3.5-T$^5$ in FLOPs and Latency. Latency is measured in one H20 GPU.

## 5.4. Ablation Study

**Full vs. Partial Weight Inheritance.** We investigate the weight inheritance strategy by training on DiT-S/2 with a batch size of 256 for 50K steps(400K steps for original DiT), where TTT-specific parameters are trained with a learning rate scaled by a factor of 2$\times$ relative to the base learning rate. We compare three inheritance schemes: inheriting only MLP weights, inheriting only attention weights, and full inheritance. Results are shown in Table 10. Unlike LiT ([Wang et al., 2025a](#)) and Diffusion Grafting ([Chandrasegaran et al., 2026](#)) which only inherit MLP weights, our linear structure exhibits strong compatibility with Softmax attention, enabling full weight inheritance while achieving performance comparable to the Softmax baseline with significantly fewer training steps. This allows our method to better leverage the knowledge encoded in pretrained Transformer weights.

**Learning Rate Strategy for TTT-Specific Parameters.** Since TTT-specific parameters are randomly initialized while other parameters are inherited from pretrained checkpoints, we propose scaling the learning rate for TTT-specific parameters by a multiplicative factor to accelerate convergence. We conduct experiments on both DiT-S/2 and DeiT, substituting attention module with proposed TTT attention,with results presented in Table 11. On DiT-S/2, using a 2$\times$ learning rate for TTT-specific parameters improves FID from 68.84 to 68.52. On DeiT, we observe improvements as the multiplier increases.These results demonstrate a higher learning rate to TTT-specific parameters is necessary. To maintain consistency, in all the image classification experiments above, we set the learning rate for newly initialized parameters to 20$\times$ the base learning rate.

**Effect of NAT.** We conduct ablation studies on the Neighbourhood Attention (NAT) component using DiT-S/2. Results are reported in Table 12. Without NAT, TTT still achieves an FID of 72.98 compared to 68.52 with NAT—a marginal gap. This indicates that NAT serves as an optional enhancement with few FLOPs added, rather than a core component of our method. Thus, in our text-to-image experiments, we omit NAT entirely and replace Softmax attention with pure TTT blocks.

## 6. Conclusion

In this work, we address Transformer linearization from a structural perspective. We identify T$^5$, a linear-complexity architecture that enables direct weight inheritance from Softmax attention. After inheriting pretrained weights, T$^5$ achieves competitive performance with only minimal fine-tuning while reducing computational complexity. Our training procedure is highly straightforward, requiring no distillation or multi-stage training strategies. We validate the effectiveness of our framework across image classification, class-conditional generation, and text-to-image generation. Notably, on SD3.5-Medium, we achieve performance comparable to the Softmax model with only 3,000 training steps while reducing computational FLOPs. We hope our work provides new insights and pathways for training large-scale models with novel architectures.

## Acknowledgements

This work is supported in part by the National Key R&D Program of China under Grant 2024YFB4708200, the National Natural Science Foundation of China under Grants U24B20173 and U2541227, and the Scientific Research Innovation Capability Support Project for Young Faculty under Grant ZYGXQNJSKYCXNLZCXM-I20.

## Impact Statement

This paper presents work whose goal is to advance the field of Machine Learning. There are many potential societal consequences of our work, none which we feel must be specifically highlighted here.

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

# A. Image Classification

We adopt the 30-epoch fine-tuning learning rate schedule from Swin Transformer (Liu et al., 2021). The detailed hyperparameters are summarized in Table 8. For linear attention, we use ELU + 1 as the activation function. For TTT, following ViT[3] (Han et al., 2026b), we set the activation function to SiLU and adopt the inner product loss for the inner model objective.

*Table 8.* Hyperparameters for image classification on ImageNet-1K.

| Hyperparameter | Value |
| --- | --- |
| Optimizer | AdamW |
| Base learning rate | $2 \times 10^{-5}$ |
| Weight decay | $1 \times 10^{-8}$ |
| Epochs | 30 |
| Learning rate schedule | Cosine decay |
| Warmup epochs | 5 |
| Warmup learning rate | $2 \times 10^{-8}$ |
| Min learning rate | $2 \times 10^{-7}$ |
| Drop path | 0.0 |
| New params lr multiplier | 20$\times$ |
| Activation (Linear Attn) | ELU+1 |
| Activation (TTT) | SiLU |
| TTT inner loss | inner product loss |

# B. Text-to-Image Generation

**Architecture.** SD3.5-Medium consists of 24 MMDiT blocks, among which the first 13 blocks are dual-attention blocks that contain both the original text-image joint attention and an additional image-token self-attention branch. In these 13 early blocks, we replace only the additional image-token self-attention branch with our proposed module, while keeping the original text-image joint attention unchanged. For the remaining standard MMDiT blocks, we replace the joint attention in the last 5 blocks. Since TTT-DWC$_{QK}$ performs depthwise convolution over neighboring tokens, we restrict the convolution to image sequences when operating within blocks that process concatenated text-image tokens.

**Training and Evaluation.** The training hyperparameters are summarized in Table 9. For DPGBench and GenEval evaluation, we set the number of inference steps to 28 and the guidance scale to 3.5. We generate 4 images per prompt for all experiments.

*Table 9.* Hyperparameters for text-to-image generation (SD3.5).

| Hyperparameter | Value |
| --- | --- |
| Optimizer | AdamW |
| Base learning rate | $1 \times 10^{-5}$ |
| Weight decay | $1 \times 10^{-4}$ |
| Batch size | 64 |
| Training steps | 3000 |
| Learning rate schedule | Constant |
| Replaced blocks | Dual (13) + Joint (last 5) |

## C. Ablation Study

*Table 10.* Ablation on weight inheritance strategies. All models are trained on DiT-S/2 with batch size 256 for 50K steps.

| Inheritance Strategy | FID↓ | IS↑ |
|---|---|---|
| Softmax Baseline (DiT-S/2) | 68.40 | – |
| MLP weights only | 89.71 | 15.84 |
| Attention weights only | 93.18 | 14.69 |
| Full inheritance | 68.52 | 20.65 |

*Table 11.* Ablation study on learning rate scaling for TTT-specific parameters. (a) DiT-S/2 results. (b) DeiT-tiny results.

| (a) DiT-S/2 | | (b) DeiT-tiny | |
|---|---|---|---|
| LR Ratio | FID↓ | LR Ratio | Top-1 Acc.↑ |
| 1× | 68.84 | 1× | 68.14 |
| 2× | **68.52** | 10× | 70.71 |
| | | 20× | 71.19 |
| | | 30× | 71.29 |
| | | 40× | **71.36** |
| | | 50× | 71.31 |
| | | 100× | 71.35 |

*Table 12.* Ablation study on Neighborhood Attention (NAT) using DiT-S/2 on ImageNet 256×256.

| NAT | FID↓ | IS↑ |
|---|---|---|
| ✓ | 68.52 | 20.65 |
| ✗ | 72.98 | 19.18 |

