# OpenReview forum: "Linearizing Vision Transformer with Test-Time Training"
_ICML.cc/2026/Conference — ICML 2026 regular_

### Official Review · Reviewer_FzwU · 2026-03-10

**Soundness:** 3
**Presentation:** 3
**Significance:** 3
**Originality:** 3
**Overall Recommendation:** 4
**Confidence:** 4

**Summary:**

This paper proposes a practical conversion pipeline that replaces $O(n^2)$ Softmax Attention in pretrained models with linear-complexity attention via Test-Time Training (TTT). The key claim is that the converted model requires only a small number of calibration iterations during inference-time adaptation while preserving a favorable efficiency-performance trade-off. The observation of substantial key bias in large-scale pretrained Transformers is particularly insightful, as it may explain why many prior transfer methods suffer from weak inheritance and training instability. In contrast to prior approaches that inherit only partial weights, this work enables full inheritance of attention-related weights.

**Compliance With Llm Reviewing Policy:**

Affirmed.

**Final Justification:**

Thanks for the response. I have no further concerns, and I would like to keep my positive score.

**Key Questions For Authors:**

1. The paper argues that Softmax Attention can be reformulated as a two-layer dynamic MLP, and that this structural correspondence enables direct weight inheritance by TTT (Eq. 4-6, Lines 168-210). However, Softmax also introduces normalized competition and probabilistic constraints over attention scores. Could the authors clarify whether TTT and Softmax Attention are theoretically equivalent in representational capacity, or only approximately aligned? Does removing Softmax normalization materially change the model's inductive bias?

2. The manuscript attributes instability in inherited TTT training primarily to Key Shift in pretrained Transformers (Lines 233-274). In practice, instability may also arise from multiple confounders, including initialization of new parameters, inner-loop gradient scaling, and mixed-precision numerical behavior. Could the authors provide stronger theoretical evidence or targeted ablations to establish that Key Shift is the principal cause rather than one factor among many?

3. The paper applies InstanceNorm to keys to remove shift and stabilize training (Table 2, Lines 248-274). Since this operation also changes pretrained key statistics, could the authors analyze whether InstanceNorm degrades inherited feature information? Are there alternatives that preserve distributional structure while still achieving shift-invariance alignment?

4. How to initialize the $W_1, W_2$? Apply the loss $\mathcal{L} _ t = \frac{1}{2}||f _ W (k _ t) - v _ t||^2$ to train $W_1, W_2$, what is the mathematical meaning of $W_1, W_2$ obtained in this way? Why can they be directly used in the functional form of $TTT(q)$? What is the mathematical principle and intuition behind it?

**Limitations:**

Although the paper includes an Impact Statement, it mainly states a broad intention to advance machine learning and does not sufficiently discuss concrete limitations or potential side effects (Lines 440-445).

Recommended additions:
1. Scope of applicability: It remains unclear whether the method transfers equally well to other domains, such as large language models or multimodal Transformers.
2. Dependence on structural assumptions: If TTT does not fully match Softmax representational capacity, successful weight inheritance may be task- or architecture-dependent.
3. Additional architectural complexity: TTT is not fully aligned with Softmax by default and requires auxiliary alignment modules.
4. Engineering and system overhead: Although inference complexity is reduced, TTT introduces inner-loop parameter updates at test time and extra modules, which may increase implementation complexity and memory overhead.

**Strengths And Weaknesses:**

Strength:
- The submission is technically well-motivated and proposes a Softmax-to-linear conversion strategy that supports full inheritance of KV-related weights, requiring only a small amount of calibration training during testing.
- The reinterpretation of Softmax Attention as a dynamic two-layer MLP provides a conceptually useful perspective for understanding the conversion mechanism.
- The observation of Key Shift (key bias) in pretrained Transformers is insightful and may help explain instability in previous transfer pipelines.
- Overall presentation quality is good; the manuscript is coherent and readable.

Weakness:
- I am not fully convinced that Softmax Attention and the proposed formulation are fundamentally equivalent. Softmax Attention relies on normalized competition induced by token-wise similarity and probability-simplex constraints on attention scores, which differ from the inductive bias of fitting $V$ from $K$ in the TTT internal model. A more rigorous theoretical justification would strengthen the claim.
- The explanation of the replacement mechanism could be expanded. For example, interpreting value prediction as compressing KV memory into a parametric functional representation may help readers better understand the method.
- It is unclear whether Key Shift is the dominant source of instability. Other factors such as initialization of newly introduced parameters, inner-loop gradient-scale dynamics, and mixed-precision numerical issues may also contribute.
While InstanceNorm improves stability, it also alters key statistics, and the paper does not sufficiently discuss whether this may lead to potential information distortion.

Presentation Bugs:
- Figure 1 appears to have a rendering or compatibility issue and may not display correctly on some non-Windows platforms(such as MacOS devices or IOS devices); the authors may need to check the image format.

---

> ### Author Rebuttal · Authors · 2026-03-31
>
> **1. Structural and Representational Similarity between Softmax and TTT**(k1)
>
> We would like to clarify that softmax attention and TTT share structural and representational similarities, but are not strictly equivalent.
> - As shown in Eq. 4, Softmax can be seen as constructing a **nonlinear dynamic MLP** using cached keys and values. Its capacity is no weaker than a 2-layer MLP, and the softmax function introduces an inductive bias of competition and probabilistic constraints.
> - TTT also constructs a **nonlinear dynamic MLP** using keys and values, with comparable nonlinear representational power. Although it may initially lack softmax's specific inductive bias, TTT can imitate softmax behavior through its internal learning process.
> - In contrast, linear attention only forms a **linear** transformation K$^{\top}$V, whose representational capacity is insufficient to approximate softmax attention.
>
> In summary, linear attention lacks the representational power to match Softmax, while TTT has the potential to bridge this gap.
>
> **2. Key Shift is the Dominant Factor behind Unstable Fine-tuning**(k2)
>
> We offer theoretical and experimental analysis to clarify the importance of key shift and our method. We denote keys $k_t, t \in [N]$ as having zero mean, and $b$ as the bias term.
>
> - First,we derive the impact of the key shift on the internal model's gradients. With a two-layer MLP as the inner model and scaled inner-product loss:
> $
> \mathcal{L}_t(\\mathbf{k_t}) = -\\frac{\\alpha}{N}\\, \\mathbf{v}_t^\\top f_W(\\mathbf{k}_t), \\quad f_W(\\mathbf{k}) = \\phi(\\mathbf{k}\\,\\mathbf{W}_1)\\,\\mathbf{W}_2
> $
>
> The gradient w.r.t. $\mathbf{W}_1$ with bias is:
>
> $
> \\nabla_{\\mathbf{W}_1} \\mathcal{L}_t(\\mathbf{k}_t+\\mathbf{b}) = -\\frac{\\alpha}{N}\\, (\\mathbf{k}_t+\\mathbf{b})^\\top \\Big[\\mathbf{v}_t\\,\\mathbf{W}_2^\\top \\odot \\phi'((\\mathbf{k}_t+\\mathbf{b})\\mathbf{W}_1)\\Big]
> $
>
> Applying a first-order Taylor expansion:
>
> $
> \\phi'\\big((\\mathbf{k}_t + \\mathbf{b})\\mathbf{W}_1\\big) \\approx \\phi'(\\mathbf{k}_t\\mathbf{W}_1) + \\phi''(\\mathbf{k}_t\\mathbf{W}_1) \\odot (\\mathbf{b}\\,\\mathbf{W}_1)
> $
>
> The gradient decomposes by orders of $\mathbf{b}$:
>
> $
> \\nabla_{\\mathbf{W}_1}\\mathcal{L}_t(\\mathbf{k}_t+\\mathbf{b}) \\approx -\\frac{\\alpha}{N} ( G_0 + G_1 + G_2 )
> $
>
> where
>
> $
> G_0 = \\mathbf{k}_t^\\top [\\mathbf{v}_t \\mathbf{W}_2^\\top \\odot \\phi'(\\mathbf{k}_t\\mathbf{W}_1)] \\quad (\\mathcal{O}(\\mathbf{b}^0))
> $
>
> $
> G_1 = \\mathbf{b}^\\top [\\mathbf{v}_t \\mathbf{W}_2^\\top \\odot \\phi'(\\mathbf{k}_t\\mathbf{W}_1)] + \\mathbf{k}_t^\\top [\\mathbf{v}_t \\mathbf{W}_2^\\top \\odot \\phi''(\\mathbf{k}_t\\mathbf{W}_1) \\odot (\\mathbf{b} \\mathbf{W}_1)] \\quad (\\mathcal{O}(\\mathbf{b}^1))
> $
>
> $
> G_2 = \\mathbf{b}^\\top [\\mathbf{v}_t \\mathbf{W}_2^\\top \\odot \\phi''(\\mathbf{k}_t\\mathbf{W}_1) \\odot (\\mathbf{b} \\mathbf{W}_1)] \\quad (\\mathcal{O}(\\mathbf{b}^2))
> $
>
> The $\mathcal{O}(\mathbf{b}^1)$ and $\mathcal{O}(\mathbf{b}^2)$ terms cannot be canceled and accumulate over TTT's online updates, causing internal gradients to explode and leading to training instability.
>
> - We further conduct quantitative ablations on components of `k_instancenorm`. Removing division-by-std while keeping mean subtraction barely affects accuracy (**71.19%** → **71.15%**). This can be seen as an alternative to remove key shift. However, removing mean subtraction while keeping division-by-std immediately causes **NaN**. This confirms that removing key shift is necessary for stable inherited-weight training.
>
> **3. Impact of InstanceNorm on Key Statistics**(k3)
>
> - Mean subtraction introduces no distributional shift. From Eq. 7, for softmax attention, $k$ and $k - k_{\text{mean}}$ produce **identical outputs**, so removing the mean changes no statistics.
>
> - Division-by-std is introduced solely to improve gradient controllability. Enabling it slightly improves accuracy from **71.15%** to **71.19%**, indicating minimal distributional impact while enhancing stability.
>
> **4. About TTT Process**(k4)
>
> - Following prior TTT work, W1 and W2 are randomly initialized.
> - To understand $\mathbf{W}_1$, $\mathbf{W}_2$ and $TTT(q)$ better, we draw an analogy to softmax attention:
>
>   - Softmax attention constructs a dynamic network losslessly by using cached $\mathbf{K}$ and $\mathbf{V}$ as two linear layers. The query $\mathbf{Q}$ then acts on this dynamic network to produce output.
>   - TTT compresses the $\mathbf{K}$ and $\mathbf{V}$ cache into a compact 2-layer MLP via gradient-based learning. The query $\mathbf{Q}$ then acts on this compressed MLP in the same manner.
>   - Thus, $\mathbf{W}_1$ and $\mathbf{W}_2$ can be viewed as a learned compression of the $\mathbf{K}, \mathbf{V}$ cache, and $TTT(q)$ enables natural interaction with cached keys and values.
>
> **5.Other Response** Thanks, we will expand the explanation of replacement and fix the potential display problems and add the limitations you mentioned about LLM, engineering overhead, etc.

---

> > ### Author Rebuttal · Reviewer_FzwU · 2026-04-04
> >
> > I appreciate the authors’ rebuttal. This analysis improves the work and addresses my concerns. The formula is rendered very poorly; I hope the author will check it before submitting.

---

> > > ### Author Response · Authors · 2026-04-07
> > >
> > > Thank you for your time and valuable comments. We will incorporate the feedback discussed in the rebuttal to further improve the paper in the revision. We apologize for the formula rendering issue and assure that all formulas and figures in the revised paper will be carefully checked to ensure proper formatting and readability across different platforms.

---

### Official Review · Reviewer_13NY · 2026-03-12

**Soundness:** 2
**Presentation:** 2
**Significance:** 2
**Originality:** 2
**Overall Recommendation:** 4
**Confidence:** 2

**Summary:**

This paper introduces a method that "linearizes" a pre-trained vision or diffusion transformer to make it more efficient at high resolutions. The method applies an instance norm to keys and a "locality-enhancement" module (a depth-wise convolution on queries and keys), to make adaptation smoother. The authors show this approach improves the efficiency in which they can turn a quadratic-complexity model into a linear one. In particular, they choose "test-time training" as their linear-complexity model, which replaces attention with a small MLP that is trained on each sample.

The authors demonstrate its effectiveness on image classification, class-conditioned diffusion, and text-conditioned diffusion

**Compliance With Llm Reviewing Policy:**

Affirmed.

**Final Justification:**

The authors have successful addressed my concerns, however I am not an expert in test-time training. Please don't weigh my score heavily in the final decision.

**Key Questions For Authors:**

As far as I understand, ViT$^3$ is trained to properly initialize the inner-loop weights (kind of like meta-learning). Whereas your T$^5$ fine-tunes a non-TTT model like a standard ViT. How are your inner-loop weights initialized? Figure 3 implies that these weights are initialized from self-attention weights. How this happens is not clear to me because then we'd need to materialize the self-attention matrix to train the inner-loop weights?

Please clarify.

**Strengths And Weaknesses:**

Strengths:
- Good motivation. It should be more efficient to fine-tune/adapt a pre-trained quadratic-complexity model into a linear-complexity model than train a new linear-complexity model from scratch.
- The method, which uses an instance norm and locality bias makes sense
- The paper is generally well-written

Weaknesses:
- Experiments and baselines. On image classification (table 4) the proposed method seems to increase FLOPS and decrease accuracy. Isn't that a poor result? Table 5 seems to be missing computation measurements like FLOPs and throughput. Since the purpose of the method is to make processing more efficient, efficiency must be fully accounted. Furthermore, the paper lacks baselines, e.g. all those mentioned in the "Linearizing Pretrained Transformers" paragraph should be benchmarked or explained away. I'd also like baselines that takes a pre-trained model and swaps in self-attention for linear attention, then fine-tunes for the same number of epochs as the proposed T$^5$. This would also require equal fine-tuning hyperparameter search for each baseline.

---

> ### Author Rebuttal · Authors · 2026-03-31
>
> **1. Clarification on DeiT and DiT Results** (W)
>
> As pointed out, our method does not show performance or efficiency advantages in initial DeiT/DiT tasks. We clarify that ***these are not weaknesses***, but ***expected outcomes of our constrained settings***:
> - Our goal is to build a highly efficient method for rapid linearization of large-scale models like SD3.5. DeiT and DiT experiments serve purely as a lightweight ***testbed*** to validate methods under constrained conditions before scaling up.
> - Given this role, we intentionally use ***short sequences*** and ***extremely quick fine-tuning***:
>   - We use standard 224×224 and 256×256 resolutions (sequence lengths only 196/256). At these lengths, the $O(N^2)$ cost of Softmax attention is not yet a bottleneck, so similar FLOPs are mathematically expected.
>   - We restrict fine-tuning to extremely short durations. Full recovery is hardly achievable, but this setup fairly reveals the relative effectiveness of different linearization methods under minimal compute.
>
> **2. True Performance and Efficiency of Our Method**(W)
>
> - On SD3.5, our method recovers full performance with only 1 hour of training on 4 H20 GPUs, with notable efficiency gains at 1K/2K resolutions—fully supporting practical effectiveness.
> - When constraints are relaxed, our method also shows clear advantages:
>   - With longer sequences, TTT's cost becomes much more favorable. DeiT and DiT(supplementary results for Table5) results are as below:
>
>    |         | 224  | 448  |  512  |  640  |  738  | 1024  |  1248  |
>   | :-----: | :--: | :--: | :---: | :---: | :---: | :---: | :----: |
>   | DeiT-T  | 1.25 | 7.13 | 10.44 | 20.56 | 32.23 | 99.76 | 204.00 |
>   | T$^5$-T | 1.34 | 5.38 | 7.02  | 10.97 | 14.51 | 28.09 | 41.72  |
>
>    | Resolution | DiT FLOPs | DiT$^5$ FLOPs | DiT Throughput (img/s) | DiT$^5$ Throughput (img/s) | DiT Para | DiT$^5$ Para |
>   | :--------: | :-----------: | :-------: | :------------------------: | :--------------------: | :----------: | :------: |
>   |    256     |      119      |   119.5   |            64.3            |          56.4          |     675M     |   679M   |
>   |    512     |      525      |    475    |            13.4            |          14.4          |      /       | / |
>   |    1024    |     2910      |   1908    |            2.0             |          3.6           |      /       | / |
>
>   - With longer fine-tuning (300 epochs on ImageNet), our method reaches 76.0% Top-1 accuracy, substantially outperforming the baseline's 72.1%.
> - In summary, DeiT/DiT experiments were not the final showcase but designed around our core goal of rapid linearization. The observed results under these constraints are expected.
>
> **3. Additional Results of Our Approach vs. Existing Methods**(W)
>
> We offer clarification regarding the baseline methods, and provide additional experiments to support our claims.
>
> - Existing methods fall into two categories:
> 	- One line, such as Diffusion Grafting, investigates training strategies for replacing softmax attention with new attention module. Our approach is **fully orthogonal** to these methods.
> 	- The other line, including CLEAR, LiT, and Hedgehog, addresses both architecture and training techniques. Most training strategies in these papers—such as distillation and two-stage training—can be readily applied to our architecture for further improvement.
> - Our work focuses on finding a linear architecture that aligns with softmax attention in the structure space. Therefore, our implementation isolates advanced training techniques and focuses on architectural alignment.
>
> For a fair comparison on architectural compatibility, we only re-implement their **architectural components** within our framework: 30 epochs of fine-tuning on ImageNet without additional training techniques. Note that their training methods can also be applied to our architecture for further improvement.  To mention, LiT does not inherit the attention weights, following its paper.
>
> **ImageNet Top-1 Accuracy (30 epochs):**
>
> |  | T$^5$(+dwc$\_{qk}$) | Linear(+dwc$\_{qk}$) | T$^5$ | Linear | LiT | CLEAR | Hedgehog |
> |---|---|---|---|---|---|---|---|
> | acc | 71.19 | 69.25 | 69.25 | 63.30 | 69.52 | 68.66 | NaN |
>
> **4. Inner Weight Initialization**(Key)
>
> We do not materialize the self-attention matrix but inherit all weights from Softmax.
>
> - TTT and Softmax share similar parameters: both have Q, K, V concepts. T$^5$ inherits all Softmax parameters including $W_Q, W_K, W_V$ projection matrices, output projection matrix, and MLP weights.
> - The difference lies in the computation: $O=\text{Softmax}(QK^{\top})V$ vs. $O=\phi(qW_1')W_2'$, where $W_1'=W_1-\nabla_{W_1} L(K,V)$, $\ W_2'=W_2-\nabla_{W_2} L(K,V)$. Here $W_1, W_2$ are learnable parametric matrices, newly initialized and tuned end-to-end. The inner weights $W_1', W_2'$ consist of two components: the learnable $W_1, W_2$ and the gradient of the inner loss $L(K,V)$ computed per image.

---

> > ### Author Rebuttal · Reviewer_13NY · 2026-04-02
> >
> > Thanks to the authors for the response. I will increase my score to "weak accept", with the relatively lack of experiments holding it back from an "accept" or "strong accept". I will keep my confidence low as I have no experience with test-time training, so I defer to the other reviews and the AC's opinion.

---

> > > ### Author Response · Authors · 2026-04-07
> > >
> > > Thank you for your time and valuable comments. We are pleased that our rebuttal has adequately addressed your major concerns. And we will incorporate the content discussed in the rebuttal to further improve the manuscript in the revision.

---

### Official Review · Reviewer_vaKb · 2026-03-16

**Soundness:** 2
**Presentation:** 3
**Significance:** 2
**Originality:** 3
**Overall Recommendation:** 4
**Confidence:** 3

**Summary:**

The paper proposes converting pretrained Vision Transformers from quadratic Softmax attention to a linear-complexity Test-Time Training (TTT) attention module by arguing that TTT is structurally closer to Softmax than standard linear attention, which enables direct weight inheritance. To make this conversion stable and accurate, the authors add instance normalization on keys to handle Softmax’s shift-invariance and depthwise convolution-based locality enhancement to better match the local bias of pretrained attention. Experiments on image classification, class-conditional generation, and text-to-image generation show mixed results overall, with the strongest evidence coming from Stable Diffusion 3.5, where the converted model achieves similar or slightly better benchmark scores while reducing latency.

**Compliance With Llm Reviewing Policy:**

Affirmed.

**Final Justification:**

My major concerns have been addressed. I can see some improvement of model efficiency in the rebuttal's results on SD3.5-T5.
But I have low confidence in reviewing the part about test-time training as I have little experience with that.

**Key Questions For Authors:**

Could the authors clarify whether the SD3.5-Medium baseline in Table 6 is also fine-tuned on the same data as SD3.5-T5? The current presentation suggests that SD3.5-T5 benefits from an extra 3,000-step adaptation stage on "open-source Flux-generated images," while it is unclear whether the baseline undergoes the same procedure. If not, the reported quality comparison may be unfair, since part of the gain could come from dataset-specific fine-tuning rather than from the proposed linearization itself.

**Limitations:**

Yes.

**Strengths And Weaknesses:**

## Strengths
- The paper’s experimental analysis of the proposed fixes after using the TTT is thorough and useful. The authors show that TTT can work in place of attention; they carefully examine why the conversion succeeds, with targeted experiments on structural choices (different TTT variants), representation alignment (InstanceNorm for key shift, depthwise convolutions for locality), and training protocol (full vs. partial inheritance, LR scaling, optional NAT). This level of ablation substantially improves the paper’s credibility and provides practical guidance for future work on post-hoc transformer linearization.
- The text-to-image experiment in Section 5.3 demonstrates both competitive quality and a practical efficiency gain: SD3.5-T5 slightly improves over SD3.5-Medium on DPG-Bench and GenEval while reducing latency by 1.32× at 1024 resolution and 1.47× at 2048 resolution. This makes a compelling case that the proposed conversion is potentially useful in the high-resolution generation regime.

## Weaknesses
- The efficiency claim is not fully convincing on image classification. On ImageNet-1K, the proposed T5 models have higher FLOPs and more parameters than the original DeiT baselines, yet still show a drop in Top-1 accuracy, especially for DeiT-S (80.24 --> 79.00 for T5-S, and 79.64 even with T5+). This makes the trade-off less compelling in the classification setting, as the method does not appear to improve either accuracy or nominal compute relative to the pretrained Softmax model, and the gap is nontrivial for the small model. The authors should better justify why this should still be viewed as a favorable linearization result.
- The class-conditional generation results do not support the efficiency claim. In Table 5, DiT$^5$-XL/2 underperforms the original DiT-XL/2 on both FID and IS, so the converted model does not preserve baseline quality. Moreover, the statement that the method uses only 0.57% of DiT’s training cost is potentially misleading, since the method inherits all weights from a pretrained DiT-XL/2 checkpoint and therefore relies on the full original DiT training cost plus an additional conversion/fine-tuning stage. This should be described as post-hoc adaptation cost, not total training cost. In addition, the paper provides no latency or throughput comparison for this experiment, making the practical efficiency gain unclear in this setting.


### Minor issues
- The presentation needs to be improved for clarity. Several important details are underspecified or introduced too late, which makes the paper harder to read than necessary. For instance, Table 1 discusses 30-epoch fine-tuning results without clearly stating the dataset.

---

> ### Author Rebuttal · Authors · 2026-03-31
>
> **1. Clarification on DeiT and DiT Results**(W1)
>
> As pointed out, our method does not show performance or efficiency advantages in initial DeiT/DiT tasks. We clarify that ***these are not weaknesses***, but ***expected outcomes of our constrained settings***:
> - Our goal is to build a highly efficient method for rapid linearization of large-scale models like SD3.5. DeiT and DiT experiments serve purely as a lightweight ***testbed*** to validate methods under constrained conditions before scaling up.
> - Given this role, we intentionally use ***short sequences*** and ***extremely quick fine-tuning***:
>   - We use standard 224×224 and 256×256 resolutions (sequence lengths only 196/256). At these lengths, the $O(N^2)$ cost of Softmax attention is not yet a bottleneck, so similar FLOPs are mathematically expected.
>   - We restrict fine-tuning to extremely short durations. Full recovery is hardly achievable, but this setup fairly reveals the relative effectiveness of different linearization methods under minimal compute.
>
> **2. True Performance and Efficiency of Our Method**(W1)
>
> - On SD3.5, our method recovers full performance with only 1 hour of training on 4 H20 GPUs, with notable efficiency gains at 1K/2K resolutions—fully supporting practical effectiveness.
> - When constraints are relaxed, our method also shows clear advantages:
>   - With longer sequences, TTT's cost becomes much more favorable. On DeiT:
>
> |         | 224  | 448  |  512  |  640  |  738  | 1024  |  1248  |
> | :-----: | :--: | :--: | :---: | :---: | :---: | :---: | :----: |
> | DeiT-T  | 1.25 | 7.13 | 10.44 | 20.56 | 32.23 | 99.76 | 204.00 |
> | T$^5$-T | 1.34 | 5.38 | 7.02  | 10.97 | 14.51 | 28.09 | 41.72  |
>
>   - With longer fine-tuning (300 epochs on ImageNet), our method reaches 76.0% Top-1 accuracy, substantially outperforming the baseline's 72.1%.
> - In summary, DeiT/DiT experiments were not the final showcase but designed around our core goal of rapid linearization. The observed results under these constraints are expected.
>
> **3. Clarification on DiT$^5$ Training Cost and Additional Results**(W2)
>
> - Our primary focus is **rapid linearization** of large pretrained models trained with enormous cost and closed-source data, so reporting fine-tuning steps directly reflects this goal.
> - Leveraging expensive pretraining is our core motivation. By designing an architecture closely aligned with softmax attention, we **fully inherit** pretrained weights to achieve low fine-tuning cost. In contrast, LiT and Hyena reinitialize new attention modules, requiring more fine-tuning epochs (280 and 28) than our 8 epochs.
> - We will list DiT and DiT$^5$/LiT/Hyena training steps in two columns (pretrain cost and linearization cost) to improve rigorousness in the revised version.
>
> Below we provide FLOPs and throughput for DiT and DiT$^5$ (not reported in LiT or Diffusion Grafting). As stated, FLOPs are similar at low resolution while TTT's advantage becomes clear at high resolution.
>
> | Resolution | DiT FLOPs | DiT$^5$ FLOPs | DiT Throughput (img/s) | DiT$^5$ Throughput (img/s) | DiT Para | DiT$^5$ Para |
> | :--------: | :-------: | :-----------: | :--------------------: | :------------------------: | :------: | :----------: |
> |    256     |    119    |     119.5     |          64.3          |            56.4            |   675M   |     679M     |
> |    512     |    525    |      475      |          13.4          |            14.4            |    /     |      /       |
> |    1024    |   2910    |     1908      |          2.0           |            3.6             |    /     |      /       |
>
> **4. Additional Fine-tuning Experiments on SD3.5**(Key)
>
> Following your suggestion, we applied the identical fine-tuning procedure to SD3.5 for 3000 steps. Performance of SD3.5 and SD3.5-T$^5$ remains comparable after the same process.
>
> **DPGBench Results:**
>
> | Model           | Global | Entity | Attribute | Relation | Other | Overall |
> | :-------------- | :----: | :----: | :-------: | :------: | :---: | :-----: |
> | SD3.5_FT        | 82.90  | 89.29  |   87.54   |  92.40   | 84.90 |  82.74  |
> | SD3.5-T$^5$\_FT | 91.51  | 90.03  |   87.84   |  92.17   | 87.13 |  84.43  |
>
> **GenEval Results:**
>
> | Model           | single | two  | count | color | position | Color attri | overall |
> | :-------------- | :----: | :--: | :---: | :---: | :------: | :---------: | :-----: |
> | SD3.5_FT        |  0.98  | 0.88 | 0.71  | 0.84  |   0.27   |    0.53     |  0.70   |
> | SD3.5-T$^5$\_FT |  1.00  | 0.88 | 0.58  | 0.88  |   0.25   |    0.55     |  0.69   |
>
> SD3.5-T$^5$ scores 84.43 vs. 82.74 on DPGBench, and 0.69 vs. 0.70 on GenEval. Our original goal in the paper is to maintain the Softmax ability after few fine-tuning, which is already satisfied by the result.
>
> **5. About Minor issues**
>
> We thank the reviewer for the suggestion, we will revise in the revision. For the 30-epoch ImageNet experiment, we used the standard ImageNet training dataset with batch size 256 per GPU on 4 GPUs.

---

> > ### Author Rebuttal · Reviewer_vaKb · 2026-04-04
> >
> > My major concerns have been addressed. But I have low confidence in reviewing the part about test-time training as I have little experience with that.

---

> > > ### Author Response · Authors · 2026-04-07
> > >
> > > Thank you for your time and valuable comments. We are pleased that our rebuttal has adequately addressed your major concerns. And we will incorporate the content discussed in the rebuttal to further improve the manuscript in the revision.

---

### Decision · Program_Chairs · 2026-04-30

**Decision:**

Accept (regular)

**Comment:**

All reviewers gave positive comments, so I recommend acceptance.